# Altered Metabolism in Glioblastoma: Myeloid-Derived Suppressor Cell (MDSC) Fitness and Tumor-Infiltrating Lymphocyte (TIL) Dysfunction

**DOI:** 10.3390/ijms22094460

**Published:** 2021-04-24

**Authors:** Natalia Di Ianni, Silvia Musio, Serena Pellegatta

**Affiliations:** Unit of Immunotherapy of Brain Tumors, Department of Molecular Neuro-Oncology, Fondazione IRCCS Istituto Neurologico Carlo Besta, Via Celoria, 11, 20133 Milan, Italy; natalia.diianni@istituto-besta.it (N.D.I.); silvia.musio@istituto-besta.it (S.M.)

**Keywords:** glioblastoma, metabolism, myeloid derived suppressor cells (MDSCs), tumor infiltrating lymphocytes (TILs)

## Abstract

The metabolism of glioblastoma (GBM), the most aggressive and lethal primary brain tumor, is flexible and adaptable to different adverse conditions, such as nutrient deprivation. Beyond glycolysis, altered lipid metabolism is implicated in GBM progression. Indeed, metabolic subtypes were recently identified based on divergent glucose and lipid metabolism. GBM is also characterized by an immunosuppressive microenvironment in which myeloid-derived suppressor cells (MDSCs) are a powerful ally of tumor cells. Increasing evidence supports the interconnection between GBM and MDSC metabolic pathways. GBM cells exert a crucial contribution to MDSC recruitment and maturation within the tumor microenvironment, where the needs of tumor-infiltrating lymphocytes (TILs) with antitumor function are completely neglected. In this review, we will discuss the unique or alternative source of energy exploited by GBM and MDSCs, exploring how deprivation of specific nutrients and accumulation of toxic byproducts can induce T-cell dysfunction. Understanding the metabolic programs of these cell components and how they impact fitness or dysfunction will be useful to improve treatment modalities, including immunotherapeutic strategies.

## 1. Introduction

Metabolic dysregulation is one of the most important hallmarks of glioblastoma (GBM) [1].

To date, metabolic reprogramming of GBM cells has only been considered a consequence of hard nutrient-restricted conditions within the tumor microenvironment. Recently, new molecular subtypes of GBM were delineated by integrating not only proteomic and genomic but also metabolomic analyses [2]. Indeed, in an elegant and meticulous new study, the use of single-cell mRNA sequencing enabled us to revise the GBM subclassification into four subtypes based on biological and metabolic characteristics, not just on genetic alterations, allowing the prediction of patient outcome. Notably, the two groups recapitulated biological/developmental functions present in the normal brain. Two other pathway-based subtypes include mitochondrial (MTC) and glycolytic/plurimetabolic (GPM) GBM. It was observed that up to 20% of GBM obtain fuel by overactive mitochondria, relying exclusively on oxidative phosphorylation (OXPHOS) for energy production. This “MTC” subgroup is associated with a better patient prognosis and may be responsive to treatments currently used in clinical trials. The GPM GBM is sustained by aerobic glycolysis, lipid synthesis and storage in lipid droplets and amino acid metabolism, and excludes mitochondrial/OXPHOS activities. This specific subtype is enriched in immune/related genes consistent with the mesenchymal subtype [3,4] and is particularly resistant to current therapies [5].

In this context, deregulated GBM metabolism and other metabolic features in the TME that are involved in immunosuppression mechanisms are implicated in T cell dysfunction.

## 2. Metabolic Flexibility of GBM Allows Adaptation in the Tumor Microenvironment

### 2.1. Glycolysis

The dominant source of energy production is glucose, one of the most available nutrients in the brain taken up by cancer cells from the tumor microenvironment. Glycolysis, one of the most primitive mechanisms for energy production, is the primary pathway exploited not only in an oxygen-limiting environment, but also under aerobic conditions by complex and fast-growing cancers, such as GBM. Cancer cells rely on glycolysis for energy production not only under low-oxygen conditions but also in the presence of oxygen, a phenomenon termed the “Warburg effect” [6,7]. The advantage of glycolysis in GBM energy production extends beyond just the initial ATP pathway, serving as a center for the rapid production of cellular energy to supply other anabolic pathways and mechanisms made possible by glycolysis [1].

Enhanced glycolytic flux, supported by the upregulation of glycolytic enzymes, is essential for tumor survival and progression and correlates positively with the invasive capacity of glioma cells. The end product of glycolysis is pyruvate, which is converted in mitochondria to Acetyl-CoA before entering the TCA cycle. In highly proliferating cancer cells in the presence of oxygen, pyruvate is converted to lactate with reduced use of the TCA cycle as a result of aerobic glycolysis, of which lactate dehydrogenase (LDH) is the major molecular mediator [8,9].

Referring to the new metabolic subclassification, GPM GBM exhibits high rates of glucose uptake and lactate production, especially when compared with MTC GBM [5].

In addition to the three main energy metabolic pathways that produce ATP—glycolysis, the tricarboxylic acid (TCA) cycle, and oxidative phosphorylation (OXPHOS)—GBM cells can use other fuel sources: glutaminolysis, the pentose-phosphate pathway (PPP), and fatty acid oxidation (FAO).

### 2.2. Glutaminolysis

Glutaminolysis catabolizes glutamine (Gln) to generate ATP, NADPH and glutathione (GSH), which contribute to the regulation of redox homeostasis; Gln is an important biosynthetic precursor of nonessential amino acids, nucleic acids, and lipids [10,11]. This metabolic pathway plays a key role in supporting the growth of several cancers; it is potentiated by HIF1α stabilization mediated by lactate to support lipogenesis under hypoxic stress. Indeed, hypoxia enhances the dependence of cells on the reductive metabolism of Gln for lipid synthesis. The isocitrate dehydrogenase (IDH)1 pathway, that in normoxia is involved in carboxylation of alpha-ketoglutarate (αKG) to isocitrate, is active even under hypoxic conditions in inducing the reductive carboxylation of Gln-derived αKG for lipogenesis [12].

Glioma cells act as “glutamine traps”, upregulating the Gln membrane importer SLC1A5, whose expression was found to be higher in primary GBM metabolically classified as GPM than in MTC GBM [5,10].

Because of the mitochondrial pyruvate starvation induced by aerobic glycolysis, cancer cells need Gln as a metabolic adaptor to fuel TCA and OXPHOS, accelerating tumor anabolism [8,9,10]. Gln catabolism is regulated by the mTOR/Myc axis, which leads to increased protein synthesis and is involved in not only supporting tumor growth and proliferation, but also regulating immune responses. MYC can coordinate glutaminolysis based on the tumor context and can also regulate Gln anabolism [13].

In general, MYC can regulate glucose, glutamine, and essential amino acid metabolism to promote fatty acid biosynthesis [14].

### 2.3. Pentose Phosphate Pathway (PPP)

The PPP is a metabolic pathway parallel to glycolysis that plays a crucial role in supporting the survival and proliferation of GBM cells [15]. The pathway generates pentose phosphate and provides nicotinamide adenine dinucleotide phosphate (NADPH), which is useful for fatty acid synthesis and is needed for cell survival under stress conditions. PPP enzyme expression was upregulated in human GBM compared to normal brain tissue. In particular, in rapidly proliferating GBM cancer stem-like cells, PPP was highly active but suppressed under acute severe hypoxic conditions, reducing proliferation, favoring the migration of GSCs, and finally switching to direct glycolysis for high protection against hypoxic damage [16]. This metabolic reprogramming may also be attributed to intermittent exposure to hypoxic conditions in the tumor microenvironment. An important connection between telomerase activation and dysregulated metabolism in GBM progression was investigated. The results obtained in in-vitro and xenograft models supported the existence of the Nrf2-TERT loop involved in preserving oxidative defense responses in GBM cells by regulating the PPP pathway and glycogen accumulation [17].

### 2.4. Fatty Acid Oxidation (FAO)

Fatty acids (FA), the main components of several lipid species, are not only obtained through direct exogenous uptake from the TME but also synthesized using nutrients, such as glucose or glutamine. How lipid metabolism is reprogrammed in GBM is still poorly described; however, it is accepted that the remodeling of lipid metabolism can include alterations in FA transport, storage of lipids as droplets and β-oxidation to generate ATP. GBM develops in a lipid-rich environment; however, the availability of environmental lipids can be relatively poor, as they are generally already engaged in normal neural elements, including myelin. Since GBM cells use lipids for many structural and energetic functions, they need to be synthesized in large amounts. The activity of de novo lipid generation is crucial for the generation of all the components necessary for the membrane constitution and for GBM growth and survival [18]. FAs are the main components of most lipids, and their generation in GBM cells requires cytosolic acetyl-CoA starting from glucose, acetate via acetyl-CoA synthetase, the TCA cycle, and citrate via ATP citrate lyase [19]. Blocking enzymes specifically required for the synthesis or elongation of FA specifically impacted tumor growth by inhibiting cell proliferation and survival [20].

The proliferation and growth of GBM are also reported to be dependent on the oxidation of fatty acids rather than their production. GBM cells can utilize FA as the main component of this oxidative metabolism, and its inhibition negatively impacted the proliferation of GBM primary cell lines and progression in a xenograft model [21]. Indeed, fatty acid-binding protein 7 (FABP7), a lipid chaperone mediating fatty acid uptake and subsequent oxidation, was upregulated in GBM and GBM stem-like cells growing as neurospheres (GBM-NS). The inhibition of FABP7 expression in GBM-NS by interfering RNA negatively impacted proliferation and infiltration capacities both in vitro and in xenograft models [22]. Notably, upon energetic stress such as glucose deprivation, GBM, which preferentially uses glucose for energy generation, can accumulate large amounts of lipid droplets consisting mainly of triglycerides, providing critical reservoirs of energy and supporting tumor survival and growth. This important study suggests that GBM cells can maintain energetic homeostasis by utilizing glucose and triglycerides/lipidic droplets. After autophagy activation, triglycerides/lipidic droplets are broken down, inducing the release of stored fatty acids entering the mitochondria for energy generation [23]. An accumulation of lipid droplets and a high content of triglycerides were also observed in primary cells derived from MTC GBM. This feature, along with lipid synthesis and storage, supports the implication of lipid metabolic activities in survival and proliferation under adverse conditions [5].

### 2.5. Hypoxia, Genetic Alterations, and Metabolic Reprogramming

Hypoxic conditions within the TME influence the metabolic choices of GBM contributing to aggressiveness and rapid progression. Metabolism alterations mainly induced by hypoxic conditions promote glycolysis upregulation by HIFs [24]. However, many molecular mechanisms and genetic alterations are influencing the nutrient utilization in GBM. Alterations in growth factors signaling can elude the nutrient utilization contributing to an uncontrolled tumor progression. It was reported that receptor tyrosine kinases (RTKs), when amplified, can alter nutrient utilization by increasing MYC expression through the PI3K-AKT-mTOR signaling [25]. Amplified EGFR impacts metabolism reprogramming by driving glycolysis and lipogenesis. mTORC2 can regulate glycolysis through AKT-independent, Myc-dependent signaling [26]. In the case of lipogenesis, FA synthesis is driven by a mechanism involving an AKT–sterol regulatory element-binding protein 1 (SREBP1) [27]. Another important signaling including the SRC proto-oncogene non-receptor tyrosine kinase is likely involved in the metabolism alteration. SRC is involved in GBM proliferation, migration, and progression. Hypoxic condition regulates SRC pathway causing radio-resistance and increasing the invasion ability of cancer cells. The SRC activity in GBM and glycolysis are supposed to be regulated by MYC, through an SRC-MYC axis [28].

An important contribution to metabolism alteration is ascribed to mutant IDH-1 and IDH2 proteins. IDH mutations are an early event in oncogenesis and are detected in 80–90% of low-grade gliomas. IDH1 and IDH2 are enzymes responsible for the interconversion of isocitrate and α-KG in cytosol and mitochondria, respectively. Mutations in IDH enzymes result in the conversion of α-KG into the oncometabolite D-2-Hydroxyglutarate (D2HG), which is implicated in metabolic dysregulation. In particular, D-2HG can inhibit BCAT1 and BCAT2, inducing glioma cells dependent on glutaminase and more sensitive to oxidative stress compared to IDH wild-type glioma cells [29]. Mutant IDH1 induces changes in lipid metabolites [30,31] and cancer cells with IDH1 mutation are more dependent on citrate and FA under hypoxia [32]. Indeed, mutant IDH1 was implicated in inducing methylation of LDHA promoter with a subsequent decrease of the gene upregulated in cancer cells and essential for glycolysis [33]. Not only does D2HG alter the metabolism of cancer cells, but it also affects the infiltration of immune cells within the TME. Mutant IDH can affect immune cell chemotaxis and correlates with fewer infiltrating T cells [34].

## 3. The Intimate Interplay between GBM and MDSCs is Detrimental to TILs

Altered metabolism promotes GBM progression and creates an immunosuppressive TME supporting specific critical immune components, such as MDSCs [35,36,37]. These cells sense the tumor microenvironment and adapt to modifications of nutrients, oxygen and inflammatory signals promoting GBM growth and progression. Their migration and accumulation in the tumor site are orchestrated by a complex signaling network of chemokines such as CCL2 and result in the suppression of both innate and adaptive immunity by inhibiting natural killer (NK) cells and antitumor immune T cells, promoting the development of regulatory T cells (Tregs) and limiting dendritic cell (DC) maturation [38,39,40].

MDSCs exhibit high glycolytic flux and preferentially use metabolic pathways for their maturation from bone marrow precursors. This consumption of carbon sources could indirectly cause suppression of effector T cells. When glycolysis is activated in response to tumor-derived factors, PPP and OXPHOS are maintained at minimum levels to ensure NADPH production [41], and MDSC proliferation is supported by glycolytic metabolites and the antioxidant phosphoenolpyruvate (PEP), which downregulate ROS production [42]. In the TME, the glycolytic pathway and metabolites play crucial roles in the modulation of MDSC fitness.

Under physiological conditions, glucose is also crucial for T cell differentiation from naïve T cells into tumor T effectors, expansion, and inflammatory function. T cells sense environmental factors and nutrients and adapt their intracellular metabolic profile to optimally meet the energy demands required for clonal expansion and effector function and to modulate inflammation [43]. Naïve T cells (TNs) are in a quiescent state and primarily use OXPHOS to produce ATP and to preserve cellular housekeeping functions. Upon antigen stimulation, activated T cells increase their appetite for glucose [44]. They favor fermentation from pyruvate to lactate even in the presence of oxygen. Aerobic glycolysis, the so-called Warburg effect also in non-tumoral cells, apparently inefficient for energy production compared to mitochondrial respiration, is instead critical in promoting expression of IFNγ in activated T cells, that can be controlled independently of the 3′ untranslated region. Specifically, it was reported that enhanced activity of lactate dehydrogenase is involved in maintaining a high concentration of acetyl-coenzyme A which increases histone acetylation and Ifng transcription [45,46].

CD8+ memory T cells rely on FAO to produce energy; effector T cells use fatty acid synthesis (FAS) for their growth and proliferation, promoting lipid generation and pro-inflammatory cytokines.

Glucose deprivation by cancer cells and, to a lesser degree, MDSCs in the TME inhibits the glucose uptake and compromises the metabolism of activated T cells. While naive T cells do not need aerobic glycolysis for survival and proliferation, activated T cells require glucose for activation and subsequent production of effector cytokines. When activated T cells are hampered from engaging glycolysis, their ability to secrete IFNγ is considerably compromised. The binding of the glycolytic protein GAPDH to the 3′-untranslated region (UTR) of IFNγ mRNA prevents efficient translation and subsequent effector function of T cells [47,48]. T cell activation and function are also compromised by the immunosuppressive activity of MDSCs, that by depleting availability of essential amino acids, such as L-arginine, L-cysteine and tryptophan, can induce downregulation of zeta chain in the TCR preventing antigen recognition [49,50].

Glucose deprivation leads to PEP insufficiency and thus blocks Ca2+ and nuclear factor of activated T cells (NFAT) signaling, which is essential after TCR stimulation for T cell activation [51]. The reduced glycolytic flux in T cells induces decreased levels of Akt activity, promoting apoptosis [52,53,54]. Glucose-deprived conditions in association with lactic acid enrichment support Treg survival and immunosuppressive function [55].

After infiltration within the TME and the tumor mass, MDSCs increase the uptake of FA mediated by the scavenger receptor CD36 (fatty acid translocase) and activate the switch from glycolysis to FAO [49] as the main source of energy for producing inhibitory cytokines and exerting their immunosuppressive role. Memory T cells rely on oxidative FA catabolism mediated by the PPAR/CPT1a axis, which is required for lipid oxidation. Both tumor cells and tumor-infiltrating MDSCs utilize lipid metabolism as a source of ATP, altering FA homeostasis and limiting availability [56]. Enhanced lipid uptake, mediated by the receptors CD36 and CD204, promotes the immunosuppressive function of MDSCs [40,57].

Many metabolite cross-feeding pathways are activated between cancer cells and MDSCs. MDSCs can adapt their metabolic pattern and trigger symbiotic metabolism with cancer cells, mimicking the mechanism of glucose-lactate shuttling in the brain from astrocytes to neurons [58]. In the context of the TME, excessive accumulation of lactate produced by GBM cells can be transported in and used by MDSCs [59] (Figure 1).

### 3.1. The Role of Hypoxia in the Function of MDSCs and TILs

Low oxygen availability (hypoxia) is a hallmark of the tumor microenvironment as a consequence of rapid cancer cell proliferation. Hypoxia plays a key role in not only supporting tumor invasion but also shaping the TME and determining the immune response. Hypoxia promotes glycolysis and increases glucose uptake by tumor cells through HIF1α upregulation and stabilization [60,61]. Stabilized HIF1α alters the function of MDSCs, promoting enhanced expression of inducible nitric oxide synthetize (iNOS) and arginase and promoting their differentiation toward a tumor-associated macrophage (TAM)-like phenotype with the ability to suppress T cell function [62,63,64]. This modulation of the fitness of MDSCs is mediated by enhanced glycolysis. In addition, tumor-associated hypoxia via HIF1α upregulates the programmed cell death ligand PD-L1 on MDSCs, a modulation crucial for T cell suppression activity [65].

The hypoxic niches in GBM are enriched in cancer stem-like cells (GSCs) and are responsible for resistance to different therapeutic approaches. The analysis of lymphocyte infiltrate in primary GBM has highlighted that TILs reside in tumor regions negative for CAIX (carbonic anhydrase IX), an endogenous hypoxia marker [66]. The role of HIF1α in promoting proliferating and immunosuppressive tumor cells and MDSCs is clear, while in modulating T cells, it is still controversial and strictly dependent on different microenvironment scenarios. Its expression promotes the differentiation of Treg Foxp3+ cells in a TGFβ-dependent manner but is also required for glycolysis in CD8+ T cell activation. Low extracellular glucose conditions make HIF1α a T cell enemy promoting the expression of LAG3, a marker of exhaustion, and CD8+ T cells can preserve their effector function, with increased release of IFN/g, granzyme b and perforin, only switching toward FA metabolism enhancing PPAR-a signaling [67,68]. Continuously severe hypoxia drives T cell exhaustion mediated by the loss of mitochondrial function and the consequent accumulation of reactive oxygen species (ROS) [69,70]. 

In a TME characterized by low oxygen and low glucose, T cells are inactivated by upregulating checkpoint receptors, such as PD-1 and CTLA-4. PD-1 is expressed on activated T cells, which engage glycolysis as a source of energy generation during differentiation into effector cells. PD-1 signal influences the metabolism of T cells by abrogating the uptake and utilization of glucose, promoting β-oxidation of endogenous FA, and driving lipolysis [53]. The fat-based metabolism induced by PD1 enables T cells to persist as long-lived cells. In contrast, CTLA-4, another important immune checkpoint, is supposed to sustain the metabolic profile of non-activated T cells, inhibiting the expression of the glucose transporter GLUT1 and hampering glycolysis without FAO enhancement [71].

The PD-L1/PD1 axis has an important impact not only on T cells but also on cancer cells and MDSCs. PD-L1, expressed on cancer cells and MDSCs [72], drives a high rate of glycolysis causing glucose depletion from the TME, and impairing T cell activation. PD-L1 blockade impacts on tumor glycolytic pathway by inhibiting the PI3K/mTOR signaling and decreasing the expression of GLUT1, with a subsequently increased availability of glucose in the TME, favoring the metabolism of activated T cells and restoring IFNγ production [73].

### 3.2. Tumor Lactate Dehydrogenase A (LDHA) and Lactate Positively Regulate MDSCs but Inhibit the T Cell Response

An HIF1-a target is LDHA, which converts pyruvate to lactate and is highly expressed in tumor cells because of its high glycolytic activity. Lactate is a protumor ally that provides a more tolerant immune environment through different mechanisms. One of these mechanisms involves MDSCs, whose frequency is significantly increased in response to lactate; in contrast, LDHA loss, depleting lactate, decreases MDSC immunosuppressive function [74]. Lactate prevents HIF1α degradation and promotes its stabilization and cytosolic accumulation, thus contributing to protumor MDSC activity. In triple-negative breast cancer, LDH-A activity is responsible for high MDSC and low T cell infiltration, promoting tumor expression of G-CSF and GM-CSF, which are important cytokines controlling MDSC development, lipid metabolism and immunosuppressive function [75]. Lactate contributes to one mechanism of radioresistance in pancreatic cancer by enhancing the tumor-promoting activity of MDSCs via the GPR81/mTOR/HIF1α/STAT3 pathway [76]. The increased lactate production promotes TME acidification mediated by the ability of monocarboxylate transporter 4 (MCT4) to export lactate, which protects glycolytic cancer cells from acid pH. Lactate is not only a metabolic waste but also an energy source for tumors. In a consolidated sharing process, nonhypoxic cancer cells import extracellular lactate through the transmembrane transporter MCT1 and metabolize it for mitochondrial oxidative metabolism [77]. Its uptake plays a crucial role in maintaining redox balance mediated by NAD+ consumption and NADH production, dampening oxidative stress during cellular anabolism. Lactate acts as a signaling molecule involved in several tumor processes: angiogenesis, promoting endothelial cell proliferation, migration and vessel assembly; and tumor cell motility and migration, inducing the expression of transforming growth factor β2 (TGF-β2) and immunosuppression [78,79,80,81].

Lactic acid has direct immunosuppressive effects contributing to tumor immune escape. High concentrations of lactate inhibit effector T cell (CTL) function through decreased proliferation and cytokine production and release, impair CTL activity and recruitment to the tumor microenvironment by blocking their motility, and promote MDSC accumulation and macrophage M2-like polarization [82,83]. Lactate suppresses the PI3K/Akt/mTOR pathway, inhibiting T cell glycolysis and therefore dampening their effector function [84,85]. Furthermore, lactate induces decreased T cell activity, preventing dendritic cell differentiation and promoting their tolerogenic phenotype and Treg development [86]. Intensive aerobic glycolysis leads activated T cells to enter a state of T cell functional inactivation or anergy, and acidification of the microenvironment results in hypoxia, which itself has immunosuppressive effects. In addition, lactate promotes the generation and maintenance of Treg Foxp3+ by stabilizing HIF1α.

### 3.3. Amino Acid Metabolism Modulates the Immune Response

Immune functions are also regulated by amino acids such as glutamine (Gln), arginine (Arg) and tryptophan (Trp). Gln, as described above, is crucial to sustaining fast-growing tumors; it is also essential for MDSC maturation and proliferation. Gln protects activated T cells from death and is necessary for modulating the antigen-mediated immune response. Gln starvation inhibits mTORC1 activity and c-myc expression, promoting the development of Tregs and inhibiting T cell activation and function. 

Arginine, regulating the expression of components of the T cell receptor, promotes T cell proliferation [58]. Arg, in the tumor environment, is metabolized by iNOS to produce nitric oxide (NO) and by arginase 1 to produce ornithine and urea [87]. High expression of iNOS is involved in the malignancy of glioma and other cancers. This enzyme has been identified not only as an effector of immunosuppression mediated by MDSCs but also as a tumor regulator of their recruitment and activation. In melanoma, this iNOS function is modulated by VEGF release in an environment of chronic inflammation [88].

Arginine limitation, mediated by overexpression of iNOS and Arg in MDSCs, inhibits cell cycle progression in the G0–G1 phase and cytokine production [89]. In addition, its deprivation blocks TCR CD3ζ expression, inhibiting antigen-specific proliferation [90]. NO generated by GBM and MDSCs has tumor-promoting and immunosuppressive functions; high concentrations of NO can promote T cell death mediated by mitochondrial dysfunction with the accumulation of ROS and reactive nitrogen species (RNS) [69,91].

Tryptophan (Trp) is an essential amino acid, and its altered metabolism is a common feature in several tumors. Trp is abnormally metabolized by both GBM cells and MDSCs, and its deprivation makes T cells susceptible to death via apoptosis [92]. GBM induces immunosuppression through upregulation of idoelamine 2,3-dioxygenase (IDO) and tryptophan 2,3-dioxygenase (TDO), enzymes that catalyze the degradation of Trp following accumulation of kynurenine pathway (KP)-derived catabolites that have immunosuppressive functions. KP metabolites inhibit the proliferation and cytotoxic activity of CD8+ T cells and promote regulatory Foxp3+ T cell development from naïve CD4+ cells mediated by the aryl hydrocarbon receptor (AhR) [93]. Microenvironmental tryptophan depletion results in downregulation of TCR-CD3 and in mTORC1 inhibition in T cells that enter a state of anergy [92,94]. The Kyn-AhR pathway has also been shown to upregulate PD-1 expression by CD8+ T cells, providing an additional immunosuppressive mechanism [95]. IDO is highly expressed by aggressive and less T cell therapy-responsive tumors, and its expression correlates positively with GBM malignancy [96]. In response to IFN-g production, it is also overexpressed in MDSCs and contributes to extracellular deprivation of Trp in the local TME and prevention of the antitumoral response [97]. In a mouse melanoma model, it was demonstrated that overexpression of IDO is associated with marked recruitment and expansion of MDSCs into the tumor microenvironment and aggressive tumor growth. Instead, IDO inhibition reduces the tumor infiltration of MDSCs and inhibits their suppressive character [98].

## 4. Conclusions

In recent years, cancer metabolism has been increasingly studied, with clear potential recognized in the identification of therapeutic targets that can be exploited by following metabolic alterations in tumor cells. Many inhibitors acting on glycolysis and OXPHOS have been considered in the setting of clinical studies. Such metabolic inhibitors would have the advantage of selectively killing cancer tissue, having no detrimental effect on normal cells, and improving the TME for antitumor T cell performance. However, targeting glucose metabolism can be challenging because of the dependence of not only cancer cells and MDSCs but also effector TILs.

Other metabolic pathways can be targeted to enhance immunity and increase T cell fitness. Very relevant and critical are some dietary modifications that affect nutrient availability in the TME and are exploited in clinical studies to slow tumor progression and improve antitumor immunity [99]. There is an important correlation between diet and immune response. Hypocaloric or ketogenic diets are particularly associated with a robust increase in CD8+ T cells [100]. These conditions can be taken into consideration to enhance the efficacy of some immunotherapeutic modalities. Indeed, the possibility of improving the metabolic capacity of T cells, such as TILs or chimeric antigen receptor (CAR) T cells, through ex-vivo manipulation before administration into patients may overcome the direct effects of tumor cells and/or barriers from the TME. 

T cell metabolism can be modulated ex vivo using interleukin (IL)-7 and IL-15 that can promote memory T cell generation thanks to their ability to increase mitochondrial biogenesis and FAO [101,102]. CAR T cells could be engineered to modulate immune checkpoint inhibitor response and to improve TME modulation [103].

These observations provide an overview of how metabolism may be manipulated to impact immunotherapeutic strategies.

## Figures and Tables

**Figure 1 ijms-22-04460-f001:**
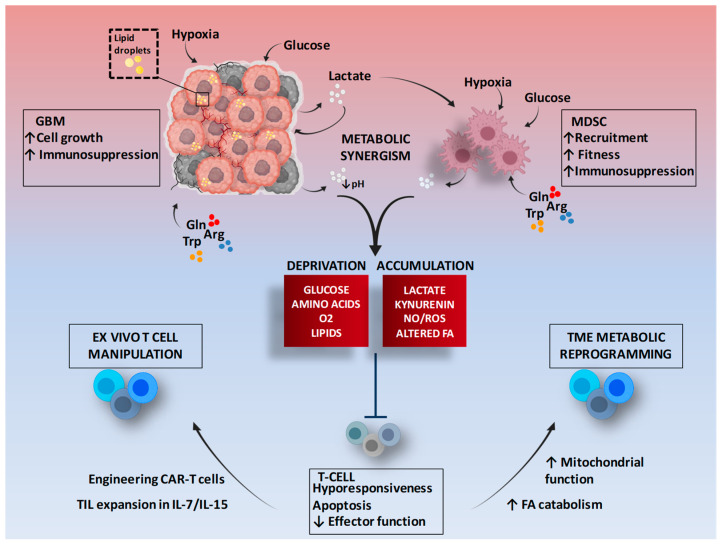
Metabolic symbiosis between GBM and MDSCs and therapeutic potential of T cell metabolic targeting. The figure shows as tumor cells and MDSCs, in a hypermetabolic state, interact to promote an immunosuppressive environment, and some mechanisms of modulation to improve the metabolic performance of T cells. Gln: Glutamine; Arg: Arginine; Trp: Tryptophan; TME: Tumor microenvironment; FA: Fatty acids; TIL: Tumor infiltrating lymphocytes; CAR: Chimeric antigen receptor. Figure adapted from images created with BioRender.com.

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
