# Peer review of "Altered Metabolism in Glioblastoma: Myeloid-Derived Suppressor Cell (MDSC) Fitness and Tumor-Infiltrating Lymphocyte (TIL) Dysfunction"

_ijms, 2021, doi:10.3390/ijms22094460_

Round 1
Reviewer 1 Report
In the review “Altered metabolism in glioblastoma: myeloid-derived suppressor cell (MDSC) fitness and infiltrating T lymphocyte (TIL) dysfunction”, the authors illustrated the metabolic rewiring in GBM, including glycolysis, glutaminolysis, pentose phosphate pathway and fatty acid oxidation. Moreovere they ​described the role of myeloid-derived suppressor cell and infiltrating T lymphocyte inside the tumor microenvironment focusing on the metabolic processes.
The work is interesting, and the topic is consistent with the urgent need to develop advanced approaches and perspectives against glioblastoma. Indeed, studies of the metabolic mechanisms, underlying GBM microenvironment and immune response, would be helpful to develop targeted therapies and to clarify the current state of art about it. I found the paper well written and articulated and therefore pleasant to read even if some minor points could be improved and the literature references in some cases need to be expanded.
Here some suggestions:
- Line 88: Glutamine is written in the abbreviated form “Gln” without the extended name the first time it is cited.
- In the title of 3.2, I suggest using the extended name of LDHA.
- Line 15-16: “The interconnection between GBM and MDSC metabolic pathways is clearly defined.”. Please re-phrase better this sentence in the abstract: do the authors mean that they provide insight into this relationship?.
- Lines 57-58: “The reasons for a GBM dependence on glycolysis under low-oxygen conditions are clear. In contrast…… are not immediately obvious.”. I suggest not to often use terms such as "clear" or "obvious" also because the concept about GBM metabolism in hypoxic condition is discussed later in the next section (3.1), otherwise it is mandatory to elucidate here, some evidence related with hypoxia and glycolysis enhancement.
- Lines 81-82: “it is potentiated by HIF1a stabilization mediated by lactate to support lipogenesis under hypoxic stress”. Could the authors describe this correlation lipogenesis/glutaminolysis in more detail?
- Lines 172-173: “Indeed, T cell activation involves aerobic glycolysis upregulation with glucose fermentation to lactate, impairing mitochondrial pyruvate oxidation”. Could the authors explain better this notion about fermentation in aerobic condition?
- Lines 177-180: “Glucose deprivation by cancer cells and, to a lesser degree, MDSCs ……….. and depleting IFN-g production due to enhanced binding of GAPDH to its mRNA, granzyme B and perforin release.” This paragraph includes a lot of information that would be better to divide and detail more, in order to better clarify, also expanding some references.
- The authors mention many times the main contribution of hypoxia in the metabolic dysregulation. I suggest to discuss other signaling pathway promoting the metabolic rewiring in GBM (for instance, cite among others: Cancers (Basel). 2020 Oct 4;12(10):2860. The Role of Hypoxia and SRC Tyrosine Kinase in Glioblastoma Invasiveness and Radioresistance).
- The final figure summarizes the main mechanisms described in the review. However, in my opinion it deserves to be improved with some of other information or, alternatively, adding an additional figure or table, to better emphasizes the conclusions.
Author Response
Response to Reviewer 1
In the review “Altered metabolism in glioblastoma: myeloid-derived suppressor cell (MDSC) fitness and infiltrating T lymphocyte (TIL) dysfunction”, the authors illustrated the metabolic rewiring in GBM, including glycolysis, glutaminolysis, pentose phosphate pathway and fatty acid oxidation. Moreover, they described the role of myeloid-derived suppressor cell and infiltrating T lymphocyte inside the tumor microenvironment focusing on the metabolic processes.
The work is interesting, and the topic is consistent with the urgent need to develop advanced approaches and perspectives against glioblastoma. Indeed, studies of the metabolic mechanisms, underlying GBM microenvironment and immune response, would be helpful to develop targeted therapies and to clarify the current state of art about it. I found the paper well written and articulated and therefore pleasant to read even if some minor points could be improved and the literature references in some cases need to be expanded. Here some suggestions:
Response. We thank the reviewer for her/his observations and suggestions, that we have addressed point-by-point.
REV. Line 88: Glutamine is written in the abbreviated form “Gln” without the extended name the first time it is cited.
Response. We have now indicated the extended name (line 87).
REV. In the title of 3.2, I suggest using the extended name of LDHA.
Response. We have added the extended name of LDHA
REV. Line 15-16: “The interconnection between GBM and MDSC metabolic pathways is clearly defined.” Please re-phrase better this sentence in the abstract: do the authors mean that they provide insight into this relationship?
Response. We have revised the sentence in the abstract
REV. Lines 57-58: “The reasons for a GBM dependence on glycolysis under low-oxygen conditions are clear. In contrast…… are not immediately obvious.”. I suggest not too often use terms such as "clear" or "obvious" also because the concept about GBM metabolism in hypoxic condition is discussed later in the next section (3.1), otherwise it is mandatory to elucidate here, some evidence related with hypoxia and glycolysis enhancement.
Response. In the revised text, we have modified the sentence (lines 58-60).
REV. Lines 81-82: “it is potentiated by HIF1a stabilization mediated by lactate to support lipogenesis under hypoxic stress”. Could the authors describe this correlation lipogenesis/glutaminolysis in more detail?
Response. As suggested by the reviewer, we have described the correlation between lipogenesis and glutaminolysis (lines 92-96).
REV. Lines 172-173: “Indeed, T cell activation involves aerobic glycolysis upregulation with glucose fermentation to lactate, impairing mitochondrial pyruvate oxidation”. Could the authors explain better this notion about fermentation in aerobic condition?
Response. In the revised text, we have explained the notion about fermentation in aerobic condition (lines 220-227).
REV. Lines 177-180: “Glucose deprivation by cancer cells and, to a lesser degree, MDSCs ……….. and depleting IFN-g production due to enhanced binding of GAPDH to its mRNA, granzyme B and perforin release.” This paragraph includes a lot of information that would be better to divide and detail more, in order to better clarify, also expanding some references.
Response. In the revised text, we have now clarified this notion and added some references (lines 232-241).
REV. The authors mention many times the main contribution of hypoxia in the metabolic dysregulation. I suggest to discuss other signaling pathway promoting the metabolic rewiring in GBM (for instance, cite among others: Cancers (Basel). 2020 Oct 4;12(10):2860.The Role of Hypoxia and SRC Tyrosine Kinase in Glioblastoma Invasiveness and Radioresistance).
Response. We thank the reviewer for this observation. We have now added a new paragraph 2.5, where we discuss hypoxia, genetic alterations, and metabolism (lines 164-193).
REV. The final figure summarizes the main mechanisms described in the review. However, in my opinion it deserves to be improved with some of other information or, alternatively, adding an additional figure or table, to better emphasizes the conclusions.
Response. We have introduced two suggestions for improving the metabolism of T cells (lines 403-406) The figure was modified accordingly.
Reviewer 2 Report
This manuscript entitled “Altered metabolism in glioblastoma: myeloid-derived suppressor cell (MDSC) fitness and in-2 filtrating T lymphocyte (TIL) dysfunction” overviews general and GBM-specific pathological metabolism in the context of interaction between GBM cells and tumor immunity. This manuscript may help future investigation for novel treatment aganst glioblastoma, however, this reviewer have some concerns that should be addressed before recommendation for publication.
(1) IDH1/2 are important factors implicated in TCA cycle. Moreover, their mutation, representativelly R132H, results in loss of function and production of 2-hydroxyglutarate and causes grade II glioma. Although IDH-mutant glioblastoma is minor vartiant compared with IDH-wildtype one, the authors cannot refrain from describing this aspect.
(2) CTLA4, PD1, PD-L1, and immune checkpoint blockade therapy shoud be addressed.
(3) Warburg effet is an important topic of cancer metabolsm, on the other hand, alternative metabolic pathways have been also reported. Surrounding knowledge of Warburg effet should be discussed more. For example, the following article may be reffered.
Duraj T et al. Beyond the Warburg Effect: Oxidative and Glycolytic Phenotypes Coexist within the Metabolic Heterogeneity of Glioblastoma. Cells 10:202 (2021).
(4) Regarding interaction between MDSC and glioblastoma, the following articles should be reffered.
Bayik D et al. Myeloid-Derived Suppressor Cell Subsets Drive Glioblastoma Growth in a Sex-Specific Manner. Cancer Discov 10:1210-1225 (2020)
Guo X et al. Glioma exosomes mediate the expansion and function of myeloid-derived suppressor cells through microRNA-29a/Hbp1 and microRNA-92a/Prkar1a pathways. Int J Cancer 144:3111-3126 (2019)
Author Response
Response to Reviewer 2
The manuscript entitled “Altered metabolism in glioblastoma: myeloid-derived suppressor cell (MDSC) fitness and in-2 filtrating T lymphocyte (TIL) dysfunction” overviews general and GBM-specific pathological metabolism in the context of interaction between GBM cells and tumor immunity. This manuscript may help future investigation for novel treatment against glioblastoma, however, this reviewer has some concerns that should be addressed before recommendation for publication. .
Response. We thank the reviewer for her/his comment, that we have address point-by-point
REV. IDH1/2 are important factors implicated in TCA cycle. Moreover, their mutation, representatively R132H, results in loss of function and production of 2-hydroxyglutarate and causes grade II glioma. Although IDH-mutant glioblastoma is minor variant compared with IDH-wild type one, the authors cannot refrain from describing this aspect.
Response. In agreement with the reviewer, we have now addressed the role of IDH mutations in altering glioma metabolism. A new paragraph: 2.5. Hypoxia, genetic alterations, and metabolic reprogramming” is now inserted in the revised text.
REV. CTLA4, PD1, PD-L1, and immune checkpoint blockade therapy should be addressed.
Response. We thank the reviewer for this suggestion. We have now addressed this notion in the revised text (lines 288-303).
REV. Warburg effect is an important topic of cancer metabolism, on the other hand, alternative metabolic pathways have been also reported. Surrounding knowledge of Warburg effect should be discussed more. For example, the following article may be referred.
Duraj T et al. Beyond the Warburg Effect: Oxidative and Glycolytic Phenotypes Coexist within the Metabolic Heterogeneity of Glioblastoma. Cells 10:202 (2021).
Response. We have added the reference, as suggested by the reviewer. In the revised text we have explain the Warburg effect for non-tumoral cells (lines 220-227).
REV. Regarding interaction between MDSC and glioblastoma, the following articles should be referred.
Bayik D et al. Myeloid-Derived Suppressor Cell Subsets Drive Glioblastoma Growth in a Sex-Specific Manner. Cancer Discov 10:1210-1225 (2020)
Guo X et al. Glioma exosomes mediate the expansion and function of myeloid-derived suppressor cells through microRNA-29a/Hbp1 and microRNA-92a/Prkar1a pathways. Int J Cancer 144:3111-3126 (2019)
Response. We thank the reviewer for suggesting new references (36 and 37 in the revised text).
Round 2
Reviewer 2 Report
This manuscript has been improved. This reviewer can recommend this review article for publication in IJMS.